



# Mid-19th-century building structure locations in Galicia and Austrian Silesia under the Habsburg Monarchy

Dominik Kaim[1], Marcin Szwagrzyk[1], Monika Dobosz[1], Mateusz Troll[1], Krzysztof Ostafin[1]

[1]Faculty of Geography and Geology, Institute of Geography and Spatial Management, Jagiellonian University, Gronostajowa 7, 30-387 Kraków, Poland

*Correspondence to*: Dominik Kaim (dominik.kaim@uj.edu.pl)

**Abstract.** We produced a reconstruction of mid-19th-century building structure locations in former Galicia and Austrian Silesia (parts of the Habsburg Monarchy), located in present-day Czechia, Poland and Ukraine and covering more than 80 000 km$^2$. Our reconstruction was based on a homogeneous series of detailed Second Military Survey maps (1:28,800), which were the result of cadastral mapping (1:2,880) generalization. The dataset consists of two kinds of building structures based on the original map legend – residential and outbuildings (mainly farm-related buildings). The dataset's accuracy was assessed quantitatively and qualitatively using independent data sources and may serve as an important input in studying long-term socio-economic processes and human-environmental interactions or as a valuable reference for continental settlement reconstructions. The dataset is available at http://dx.doi.org/10.17632/md8jp9ny9z.1 (Kaim et al., 2020).

## 1 Introduction

Although the human impact on Earth has been ongoing for millennia (Stephens et al., 2019), it has accelerated since the mid-19th century with the development of industry, transport infrastructure and land use change (Fischer-Kowalski et al., 2014). In many regions of Europe, this has been a time of minimal forest cover due to high use from both agriculture and industry (Gingrich et al., 2019; Jepsen et al., 2015). Although there have been many land use reconstructions covering this period, they have usually focused on the dominant land uses (Fuchs et al., 2013; Lieskovský et al., 2018) or, if they are global, have offered a generalized view of settlements (Hurtt et al., 2011; Klein Goldewijk et al., 2010). Detailed, large-scale historical settlement data are either missing or highly uncertain (Lieskovský et al., 2018). Only recently have large-scale, long-term, and highly accurate settlement reconstructions become available to scholars (Leyk and Uhl, 2018). As human impacts on the landscape may result in long-lasting legacies (Fuchs et al., 2016; Munteanu et al., 2017), high-quality, spatially explicit historical settlement data need to be produced and shared. In the past, housing structures impacted the appearance of invasive species (Gavier-Pizarro et al., 2010) and increased the demand for forest litter, resulting in reduced soil carbon pools (Gimmi et al., 2013) or triggered the development and persistence of the wildland-urban interface over time (Kaim et al., 2018). These examples show that the existence of easily accessible, high-quality data on historical settlements may contribute to a better understanding of human impacts on the environment in the future.

In this paper, we introduce a dataset covering more than 1.3 million[1] building structure locations as of the mid-19th century, detected in parts of current Poland, Ukraine and Czechia, which were formerly parts of the Habsburg Monarchy (Austrian Empire). The dataset contains exact locations of residential and farm buildings in a territory covering more than 80 000 km$^2$. Our database captures the situation just before rapid industrialization (Frank, 2005), massive inter-continental migration (Praszalowicz, 2003) and profound land use changes, being a result of societal and political changes in the region (Munteanu

et al., 2014). It may be used as a stand-alone dataset for a variety of human-related analyses in the environmental and social sciences or as reference data for broad-scale (i.e., continental) reconstructions.

## 2 Dataset

### 2.1 Study area

The data were collected for parts of current Poland, Ukraine and Czechia that belonged to the Habsburg Monarchy (Austrian

Empire) in the mid-19th century. These areas were called Austrian Silesia and Galicia at the time (Figure 1). Austrian Silesia (more than 80% of its area in present-day Czechia and less than 20% in Poland) was the small southernmost part of the Silesia region, and it remained in the Habsburg Monarchy after Silesia's division in 1742. It consisted of two historical parts – Tesin Silesia and Opava Silesia – where Opava (Troppau) was the largest city (16,608 inh., 1869; Bevölkerung..., 1871). Galicia was an Austrian name introduced for part of the Crown of the Kingdom of Poland territory, when it was annexed by Austria in

1772 (~ 40% of its area is in present-day Poland and the rest is in Ukraine)[2]. Galicia was one of the largest and most populated crownlands in the Austrian Empire and Austria-Hungary, and agriculture dominated its economy. Two prominent cities in Galicia were Lviv (87,109 inh., 1869) and Kraków (49,835 inh., 1869; Bevölkerung..., 1871).

### 2.2 Materials and methods

#### 2.2.1 Historical maps

The reconstruction was based on a homogeneous set of Second Military Survey maps, which were acquired from the War Archive in Vienna in the form of scanned .tif files (at 300 DPI). The maps for Austrian Silesia (42 map sheets) were published in the period 1837–1841 and those for Galicia (412 map sheets) were published in the period 1861–1864. One map sheet located in the north-eastern part of Galicia was not available in the archive and could not be used in the study. The scale of the

map is 1:28,800, and it was produced as a result of generalization and update of cadastral maps (1:2,880) for military purposes (Konias, 2000). Cadastral maps were prepared in the periods 1824–1830, 1833–1836 (Silesia) and 1824–1830, 1844–1854

---

[1] The number of buildings within Galicia and Austrian Silesia was 1 305 233; however, the database also covers additional structures located on the same map sheets, thus yielding 1 327 466 structures in total.
[2] The territory annexed in 1772 was enlarged in 1775 (Bukovina), reduced in 1815 (Zamość district), enlarged in 1846 (Free City of Cracow) and reduced again in 1849 (Bukovina as a separate Austrian crownland).

(Galicia). The Second Military Survey was the first empire-wide topographic mapping initiative based on a proper map projection (Affek, 2015; Skaloš et al., 2011; Timár et al., 2010). Due to the high quality of the maps, their relatively low positional errors and their large catalogue of land use categories, they are often used in land use reconstructions in different
parts of the Habsburg Empire (Feurdean et al., 2017; Kaim et al., 2016; Munteanu et al., 2015; Pavelková et al., 2016).

### 2.2.2 Building images in the Second Military Survey

Although the map scales of cadastral mapping and military mapping differ substantially, the images of buildings on the Second Military Survey are detailed (Figure 2). However, to assess the differences between the maps, we first conducted a systematic
comparison between the maps by comparing the building information presented by the Second Military Survey to that in the cadastral maps. The procedure aimed at assessing the impact of map generalization on the potential number of structures that we could acquire. We selected ten case study areas located in the different parts of Galicia (eight cases) and Austrian Silesia (two cases) representing different landscape conditions. The selection was determined by the availability of cadastral maps, which was a true obstacle, especially for the Galician part of the study area. Finally, we used resources available at
www.szukajwarchiwach.gov.pl, an official website for documents stored in the Polish national archives, and www.geshergalicia.org, a nonprofit organization supporting Jewish genealogical and historical research on Galicia. Maps presented on the website were originally stored in national archives in Poland and Ukraine. Cadastral maps from Austrian Silesia were consulted on the website of the State Administration of Land Surveying and Cadastre of Czechia at https://archivnimapy.cuzk.cz. We selected easily identifiable parts of villages and towns, counting the building structures on
the cadastral maps and compared them to the Second Military Survey maps (Table 1). We found that although the structure images in the Second Military Survey maps are very detailed, historical cartographers had to employ generalization procedures. On average, the number of buildings presented on the Second Military Survey maps was nearly 85% of the number of buildings presented on the cadastral maps (Table 1); however, the results in towns were lower and those in rural areas were higher due to the generalization procedures. Despite the differences, we decided that the building structures are presented with very high
quality, which makes it possible to obtain reasonably accurate structures (Figure 2).

### 2.2.3 Geometric correction and georeferencing

Different referenced data were employed to georeference the maps. In the case of the Polish part of Austrian Silesia and Galicia, Polish topographic maps from the 1970s at a scale of 1:25 000 were used. Maps elaborated in the Polish 1965
coordinate system based on the Pulkovo-42 reference frame were obtained as raster images transformed to the PL-1992 coordinate system based on the ETRF-89 reference frame (explanations on terms used in this paragraph can be found in Appendix A). In the case of the Ukrainian part of the Galicia, high-resolution World Imagery and DigitalGlobe imagery as well as Soviet military topographic maps at scales of 1:25 000, 1:50 000 and 1:100 000 were used. Soviet maps elaborated in
the 1942 coordinate system based on the Pulkovo-42 reference frame were transformed to the proper zone of the UTM
coordinate system. In the case of the Czech part of Austrian Silesia, the local-level administrative boundaries were used as
georeferenced information.

Original map sheets from the Second Military Survey were cropped along the map frame. Each cropped image was processed
separately using at least 20 control points per sheet. Points were chosen from triangulation points, historical buildings (e.g.,
churches), recognizable cross-roads, bridges, viaducts and local administrative boundaries. If such points were lacking,
river/stream connections were also used. Geometric correction and georeferencing to the PL-1992 or UTM coordinate system
was obtained using 2nd-order polynomial transformation. For map sheets with low coverage along the borderland, a 1st-order
polynomial transformation was applied. The total RMS Error (Root Mean Square Error) for most sheets reached values
between 10 and 30 metres, occasionally exceeding 30 metres, which indicates the level of geometric accuracy of the final
dataset.


### 2.2.4 Building structure acquisition

The maps show two main categories of buildings in different colours – red (ger. *Wohngebäude*), indicating mainly residential
buildings (but also including some churches, monasteries, town halls or railway stations; in total, approximately 1% of the
'red' buildings were non-residential) and black (ger. *Wirtschaftsgebäude*), including farm or agriculture-related buildings (but
also including similar exceptions mentioned above) (Zaffauk, 1889). To be consistent with the map content, we decided to
acquire the structures according to these two categories. The division was not related to the materials used to construct the
building structure (wood, bricks, stones), as it was presented on the cadastral maps, apart from the black structures surrounded
by the red border, which meant outbuildings built of stone or brick.

We used a semiautomatic, colour-based method involving the classification toolbar from ArcMap software to acquire
residential buildings (Figure 3) and manual vectorization for farm buildings. In the first step, the training data were manually
digitized for 12 randomly selected map sheets. The training polygons included two classes: buildings and all the other objects.
Based on the training data, signatures for red, green and blue raster bands were produced and used for map sheet classification.
Classified raster images were then converted into vectors and cleaned. The cleaning of vectors classified as buildings included
removing small and irregularly shaped polygons or other shapes based on their area. The procedure described above was then
performed for all of the map sheets using a loop. After the semiautomatic procedure, each map sheet was verified manually to
eliminate commissions and omissions. As black was a widespread colour on the map, we decided to acquire all the farm
buildings by manual vectorization, as visually inspecting the errors was a time-consuming process. All the buildings were
finally combined into one layer and attributed a function – residential (originally red) or farm (originally black). Additionally,
we assigned each building a date based on when the map sheet was published. The final layer was transferred to the LAEA
coordinate system.



## 3 Technical validation

The data presented in the paper were subject to several accuracy assessments. We assessed the acquisition accuracy and referred the data with the census data at different administrative levels. Additionally, we verified the number of buildings with the textual information presented on the original map sheets in the form of building number summaries and used auxiliary data
such as cadastral maps as needed.

### 3.1 Relation between mapped structures and vectorized structures

The relation between the structures presented on the Second Military Survey maps and structures captured in our database was assessed by comparing the numbers of both values in randomly selected, nonoverlapping circles (300 m ratio; area – 28.27 ha)
located across the study area. First, we selected 300 circles, verified them visually and counted the structures found on the map. Then, we decided to remove from the next steps of the comparison those circles in places where there were no buildings either on the map or in the database. The final number of test circles was thus reduced to 93. After comparing the number of structures, we calculated the Root Mean Square Error (RMSE) and the correlations (Pearson's $r$) between the structures' sums on the map and in the database for all 93 test areas. The RMSE was based on the formula (Eq. 1):

$$RMSE = \sqrt{\sum_{i=1}^{n} \frac{(P_i - O_i)^2}{n}} \qquad (1)$$

where:

$P_i$ – predicted values (vectorized building structures)

$O_i$ – observed values (building structures on the maps)

$n$ – sample size (number of test areas)


The procedure was employed for three conditions: residential buildings, farm-related buildings and all buildings. Additionally, the results of the accuracy assessment were represented by a confusion matrix comparing user's accuracy, producer's accuracy, overall accuracy, the kappa coefficient of agreement and the F score, which is a harmonic mean of producer's and user's accuracy (Fawcett, 2006; Leyk and Uhl, 2018).
The results show that the numbers of buildings present on the maps were very similar to the numbers we acquired. The RMSE values were equal to 1.92 for the condition with all buildings, 1.55 for the condition with only residential buildings and 1.53 for the condition with only farm-related buildings (the mean values of structures found in the test circles on the maps were 17, 12 and 6, respectively). The correlations between the structures presented on the maps and the buildings we acquired were also very high – it was $r = 0.997$ for the total number of buildings, $r = 0.996$ for residential buildings, and $r = 0.987$ for farm-related
structures (Figure 4).





The overall accuracy for all buildings we acquired was 93.65%; however, it was higher for residential buildings (96.98%) than for farm-related structures (95.53%). Similarly, the slightly higher quality of the residential building class was supported by the F score (Table 2). The kappa coefficient for the classification procedure was 0.86.

### 3.2 Completeness - reference to census data on the district level

To verify the total number of houses acquired in our procedure for the independent source, we compared the number of vectorized structures with the information from the census data at the district level for the whole study area (n=99). The censuses closest in time to the publication of the maps were organized in 1857 for Austrian Silesia (n=23) and in 1869 for Galicia (n=76). Although there is a time difference between the maps and the census data (~18 years in Austrian Silesia and ~7 years in Galicia), there was no better option for comparing the number of buildings for those regions due to the timing of the census. Additionally, we could verify in the sources only the number of residential buildings (which account for 69% of our structures), as the census did not contain information on farm-related structures.

The results show that the number of houses recorded in the census data and captured in our database differed; however, the difference was not great. While the censuses indicated 914 107 structures, we acquired 897 020 buildings of a similar type for the whole study area. However, regionally, the differences were diverse. Comparison at the district level indicated that on average, we acquired 99.4% of the houses recorded by the census data, but the differences among the districts were substantial (Figure 5). We found that the number of vectorized residential buildings for the districts located in Austrian Silesia was usually higher than the number recorded in the census. At the same time, in Galicia, the differences were wider ranging, as both overestimations and underestimations could be found. In two districts (Żółkiew, *ukr.* Жовква and Staremiasto (*Staryj Sambir*, *ukr.* Старий Самбір)) the number of houses we vectorized was less than 70% of the houses recorded by the census. What is interesting, however, is that if we compare all the structures we acquired from the maps (residential and farm-related buildings together), their sum accounts for 99.4% and 98.9% of the houses recorded in the census for these districts, respectively. This may suggest that the building division, as presented on the map, might have been understood in a different way by different cartographers. However, this hypothesis can only be confirmed through additional research and deeper study, only partly conducted within this paper (see below, section 3.4). Unfortunately, original map instructions for the Second Military Survey are not available in the archives and could not be consulted.

### 3.3 Completeness - reference to map frame information

Each map sheet (approx. 15x15 km) of the Second Military Survey has additional textual information on the frame, where basic statistics, important from a military point of view, are presented. The statistics, usually presented at the village level, include the number of houses, number of stables and number of people and horses that could be stationed there. We used this information to verify the number of houses we captured in the database by choosing 10 evenly distributed map sheets (2 from

Austrian Silesia, 4 from the western part of Galicia, 4 from the eastern part of Galicia; Figure 6A) representing different landscape conditions (lowlands, foothills, mountainous areas) and comparing the number of vectorized building structures

within them at the village level (Figure 6B). Since the number of stables was not fully comparable with the number of farm buildings in our database, we decided to compare the number of houses only. In some cases, the villages were split into neighbouring map sheets, and corrections, including adding or removing some buildings located within the specified villages, had to be implemented (Figure 6C). In two cases, however, we found that the number of houses for the village was not listed, as two neighbouring map sheets each informed that the information was available on the other sheet. Altogether, information

from 283 towns and villages on the 10 selected map sheets was summarized.

The comparison showed that in most cases (7 out of 10), the number of houses we captured in the database was higher than the number presented on the map frame. The differences ranged from 0.4% to 54.7%, with an average difference of 14% (Table 3). A more detailed explanation of the potential reasons for this is partly presented in section 3.4, where the local-level analyses are presented.


### 3.4. Completeness - reference to census data and map sheet information on the local level

The comparisons with census data at the district level and map frame information at the map sheet level showed that while on average, our database captured information on houses relatively well, local differences were substantial. To better understand the nature and potential explanations of local differences, we present a few situations below where we deal with

underestimation or overestimation between our structures and the reference data (Table 4).

### 3.4.1. Underestimation

The analyses at the district level showed that in extreme cases, the number of houses we covered in the database was more than 30% lower than the number in the census data. We chose two villages – Jaworki and Milcza – to analyse in detail as

examples. In both cases, the differences were substantial; in Jaworki, we captured slightly more than 70.5% of the number of houses in the census and in Milcza, we captured 43% of the number of houses in the census data.

The example of Jaworki shows that the map frame information gave very similar values to those presented in the census. At the same time, the map frame provided information about the relatively low number of people that could be housed there. The number is very low when compared to that in other villages we analysed, although some of the villages were located in similar

mountainous conditions. The ethnological research performed in the village in the first half of the XX century confirmed a very low standard of living there, even in comparison to that in neighbouring areas (Reinfuss, 1947). Jaworki had an unusual system of seasonal farm buildings located higher in the mountains that were inhabited by shepherds in the summer season. Our data show that in the village, the number of farm-related structures was even higher than the number of houses, which is





also unusual. We hypothesize that some of the inhabited buildings were classified as farm-related on the map but were

residential in reality. This could explain the difference between our data and the census data.

In Milcza, both the census data and the map frame information confirmed a much higher number of houses than we captured in the database. However, in this case, the percentage of buildings in the database was even lower than the values observed in extreme situations at the district level (Figure 5), as we captured only 43% of the number of buildings in the census. Since the map frame information confirmed the values from the census, which were substantially different from those in our database,

we decided to consult the original cadastral maps to compare them with the Second Military Survey maps. The comparison showed that while the cadastral maps (1851) indicated 99 buildings, the Second Military Survey maps (1861/62) showed a total of only 49 buildings, including 37 residential buildings. This confirms an unprecedented level of map generalization here when compared to that in other areas (Table 1; Figure 7). It also explains the level of underestimation we noticed in the database when compared to other, independent sources.


### 3.4.2. Overestimation

The analysis conducted for the village of Milówka, located in the western part of the Carpathians (Western Galicia), showed that our database captured a slightly higher number of structures than that indicated in the 1869 census, and it captured a substantially higher number than that indicated in the map frame summary (Table 3). In the map frame, however, Milówka

also contained the hamlet Sucha Góra, which formally belonged to the main village, although the statistics for the hamlets were kept separately in some cases, potentially for strategic reasons. The census data were published for the commune level, and only some of the hamlets were indicated separately. Adding the numbers from the main village and the hamlet together makes the difference between our database and the census versus map frame statistics substantially smaller (Table 3). Potentially, the hamlets could have been moved from one village to another over time, which, in some cases, makes

comparisons over longer periods difficult (Ostafin et al., 2020).

In the same region, relatively close to Milówka (< 15 km), we also found that compared to the data in the census and map frame information, our dataset substantially overestimated the number of houses in the village of Trzebinia (our data had more than 170% the number listed in the 1869 census). Although we consulted cadastral maps (1844: 169 buildings, including houses) and the 1880 census data (89 houses) and verified the potential administrative boundary changes, we could not find

any objective reason for such a large difference. We must bear in mind that the mid-XIX century was a time of dramatic political movements, natural disasters, diseases, and famine in Galicia, which resulted in the most dramatic population decrease in over 100 years (Zamorski, 1989). It is hard to determine whether these events were responsible for the reduction in the number of houses over such a short period of time. It is also beyond the scope of the data descriptor to explain the socio-economic background in detail on the local level. Although the differences we observed were on average much lower than in

this extreme case, we wanted to provide this example to show potential database users that such situations are possible.



## 4 Data availability

The dataset is available at http://dx.doi.org/10.17632/md8jp9ny9z.1 (Kaim et al., 2020).

The data are stored in open, widely used shapefile (shp) format, which may be opened in GIS software (incl. open-source, like QGIS). Shapefile format consist of three mandatory files (.shp, .shx, .dbf), and the set of non-mandatory files. In case of our

file the complete set of files include:

buildings_GASID.shp – the feature geometry

buildings_GASID.shx – a positional index of the feature geometry

buildings_GASID.dbf – attribute format in dBase file

buildings_GASID.prj – projection description with text representation of coordinate reference systems

buildings_GASID.sbn, buildings_GASID.sbx – a spatial indexes of the features

buildings_GASID.xml – geospatial metadata in XML format

buildings_GASID.cpg – used to specify the code page (only for dbf) for identifying the character encoding to be used.

The files are compressed in .7z format and may be unpacked using e.g. 7-Zip https://www.7-zip.org/.

The attributes available within the dataset are:

type – the type of the building (1 - residential, 2 - farm-related)

Year 1, Year 2 – map sheet production period

comment – if map sheet production dates were not specified, we analysed the dates of neighbouring sheets and added it here as the most probable period

## 265   5 Conclusions

The data descriptor presents the complete coverage of the mid-19th century building structure locations in the historical regions of Galicia and Austrian Silesia in Central Europe. The dataset covers more than 1.3 million objects, including houses and farm-related buildings. This is the first such large and detailed database in the region. The dataset is based on the Second Military Survey maps (1:28,800), which were the result of a cadastral mapping (1:2,880) generalization for military purposes and thus

offered a much higher level of detail than earlier (e.g., the First Military Survey, 1:28,800) or later (e.g., BW editions of the Third Military Survey, 1:25,000) mapping sources in the area. This is also the only source of information on the number and location of farm-related structures at the time, as they were not covered by other, independent datasets.

The technical validation of the database showed a high level of object completeness when compared to different independent sources. Nevertheless, there were some discrepancies in the number of houses we acquired and the number according in the

census data, map frame information and cadastral mapping. We tried, however, to explain the types and reasons for the potential differences. Considering the size of the study area and number of structures we acquired, local differences cannot be



explained here, as they go beyond the scope of the data descriptor. We hope, however, that making this dataset available will enable further analysis and improve knowledge of the differences among the datasets.

**Author contribution**

DK, MS, MT and KO conceived of and designed the research; DK, MS, MD, MT and KO acquired the data; DK performed the accuracy assessment; DK wrote the paper; and DK, MS, MD, MT and KO revised the manuscript.

**Competing interests**

The authors declare no conflicts of interest.

**Acknowledgements**

This research was funded by the Ministry of Science and Higher Education, Republic of Poland under the frame of 'National Programme for the Development of Humanities' 2015–2020, as a part of the GASID project (Galicia and Austrian Silesia Interactive Database 1857–1910, 1aH 15 0324 83).

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



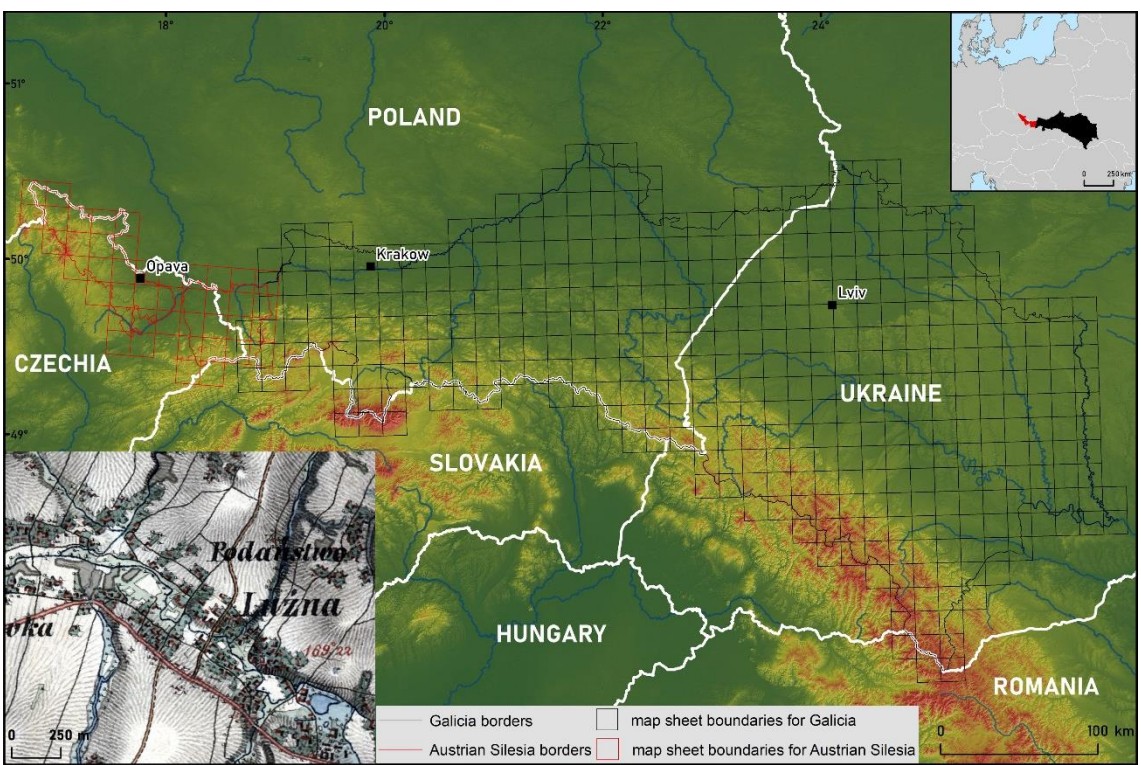

**Figure 1: Study area; lower left corner presents a small portion of the maps used in the study (source: War Archive Vienna).**


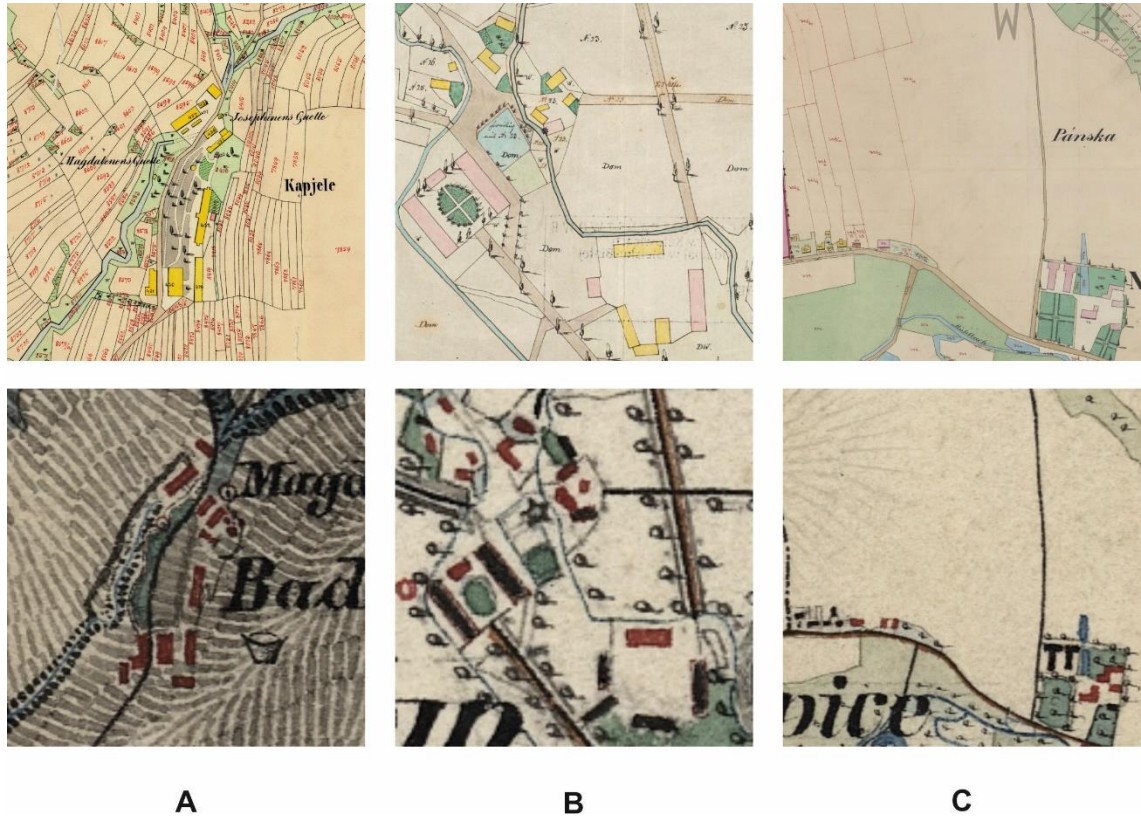

**Figure 2: Comparison of cadastral maps 1:2,880 (upper row) and Second Military Survey maps 1:28,800 (lower row) for Szczawnica (A), Kaniów (B) and Niewiarów (C) in Galicia. Source: National Archive in Kraków, National Archive in Katowice, War Archive Vienna.**




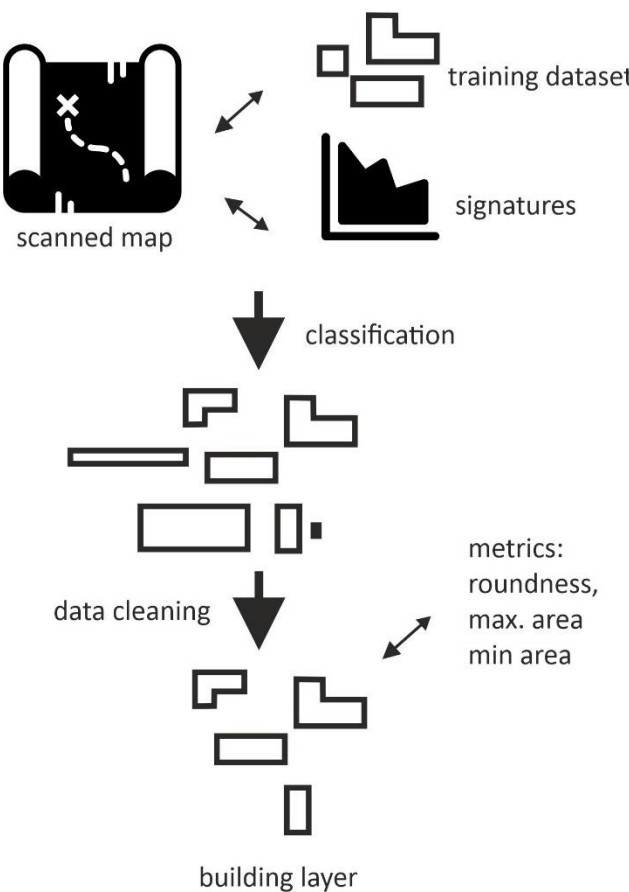

**Figure 3: Semiautomatic procedure for acquiring residential building structures (originally marked with red) from historical maps.**




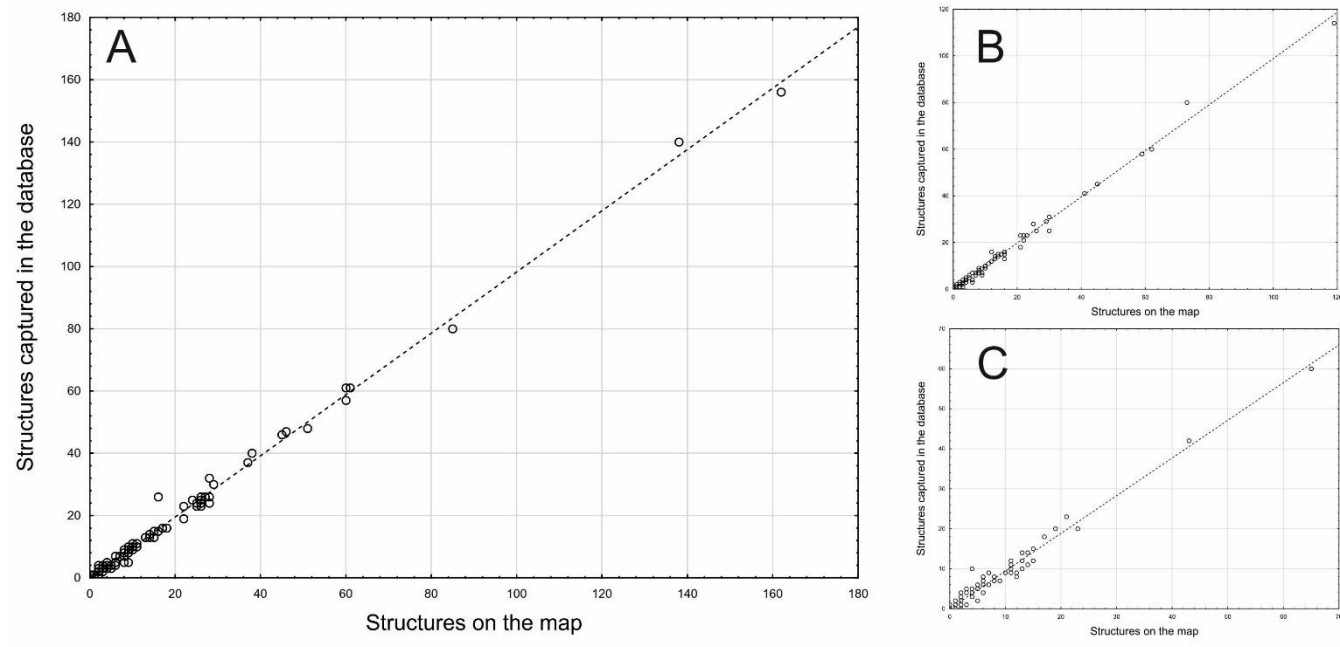

**Figure 4: Pearson's correlations *r* between the number of buildings shown on the maps and the number of structures acquired in the dataset: for all buildings (A), residential buildings (B) and farm-related buildings (C), n=93.**







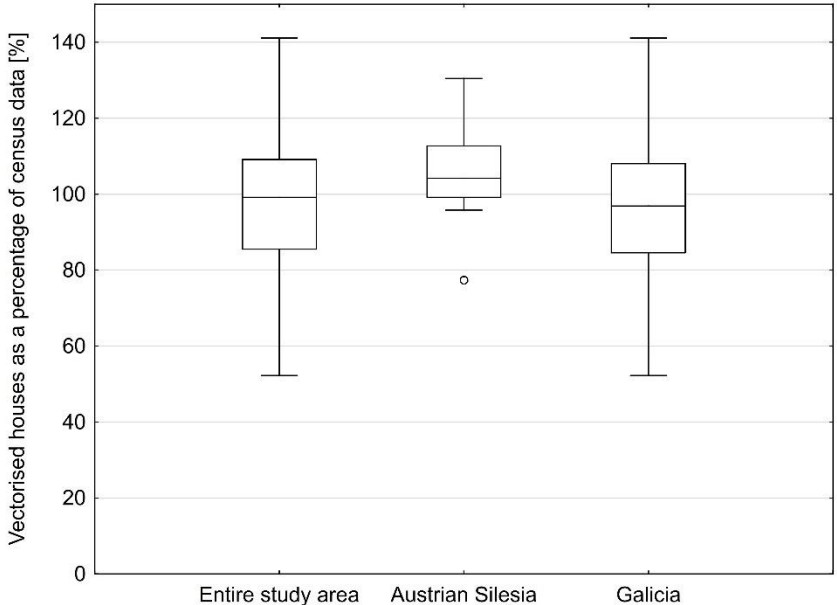

**Figure 5: Number of vectorized residential structures compared to the census data at the district level for the Entire study area (n = 99), Austrian Silesia (n=23) and Galicia (n=76).**






**Figure 6: Verification of the building structure numbers acquired from the maps with map frame information. Location of 10 evenly distributed map sheets (A); building locations and map frame information (B); information** 
**noting that statistics for the selected villages are available on the neighbouring map sheet (C); source: War Archive Vienna.**



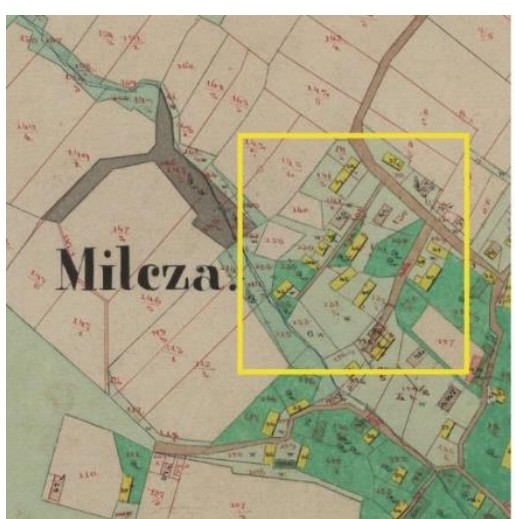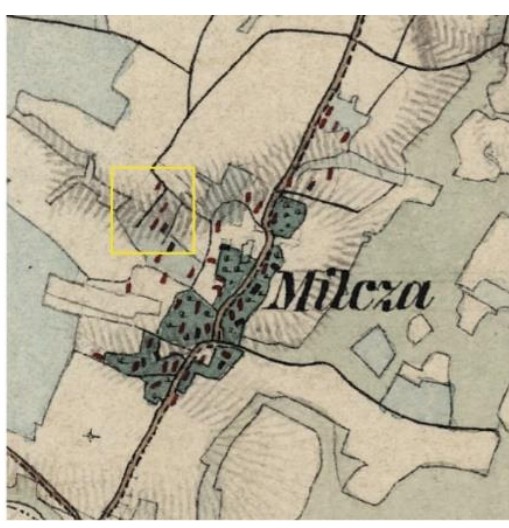

**Figure 7: Comparison of a cadastral map (1851) and Second Military Survey map (1861/62) indicating a very high level of map generalization; source: National Archive in Przemyśl, War Archive Vienna.**







**Table 1 Comparison of the number of building structures presented on the cadastral maps (1:2880) and Second Military Survey maps (1:28 800) in ten selected test areas.**


| Village/town | Cadastral maps | Second Military Survey | Second Military Survey structures as percentage of cadastral map [%] | Date of publication of cadastral map | Second Military Survey date of publication | Cadastral map link | Second Military Survey link |
|---|---|---|---|---|---|---|---|
| **Austrian Silesia** | | | | | | | |
| Krnov (Jägerndorf) | 25 | 18 | 72.0 | 1836 | 1840/41 | link | link |
| Melc (Meltsch) | 113 | 90 | 79.6 | 1836 | 1839/40 | link | link |
| **Galicia – western part** | | | | | | | |
| Baranów Sandomierski | 47 | 41 | 87.2 | 1850 | 1861/62 | link | link |
| Biecz | 39 | 34 | 87.2 | 1850 | 1861/62 | link | link |
| Gorlice | 58 | 45 | 77.6 | 1850 | 1861/62 | link | link |
| Kolbuszowa | 32 | 32 | 100.0 | 1850 | 1861/62 | link | link |
| **Galicia – eastern part** | | | | | | | |
| Jagielnica (Ягільниця) | 143 | 116 | 81.1 | 1861 | 1862/63 | link | link |
| Jaryczów Nowy (Новий Яричів) | 24 | 20 | 83.3 | 1850 | 1862/63 | link | link |
| Kulików (Куликів) | 88 | 80 | 90.9 | 1854 | 1862/63 | link | link |
| Żurów (Журів) | 43 | 38 | 88.4 | 1848 | 1862/63 | link | link |
| *Average* | | | *84.7* | | | | |

**Table 2 Accuracy assessment measures**

| | User's accuracy | Producer's accuracy | Overall accuracy | F score |
|---|---|---|---|---|
| **Residential buildings** | 98.61% | 96.81% | 96.98% | 0.98 |
| **Farm-related buildings** | 87.01% | 99.59% | 95.53% | 0.93 |




**Table 3 Comparison of the number of houses with the statistics presented on the map frames (number of houses according to the map frame = 100%)**

| Map sheet | Vectorized housed | Houses according to the map frame | Difference [%] |
|---|---|---|---|
| **Austrian Silesia** | | | |
| **Section 3 Column 11 E** | 3021 | 2276 | 32.7 |
| **Section 3 Column 6 E** | 3624 | 3637 | 0.4 |
| **Galicia – Western part** | | | |
| **Section 10 Column 11 W** | 2757 | 3818 | 27.8 |
| **Section 5 Column 13 W** | 2113 | 2828 | 25.3 |
| **Section 9 Column 23 W** | 5072 | 3279 | 54.7 |
| **Section 8 Column 18 W** | 2981 | 2632 | 13.3 |
| **Galicia – Eastern part** | | | |
| **Section 8 Column 3 E** | 3207 | 2542 | 26.2 |
| **Section 16 Column 2 W** | 66 | 46 | 43.5 |
| **Section 16 Column 8 E** | 2939 | 2848 | 3.2 |
| **Section 8 Column 6 W** | 3538 | 2953 | 19.8 |
| *Average* | | | *14* |









**Table 4. Number of houses captured in our database related to census data and map frame information – examples of underestimation and overestimation. \* - summary of two cadastral villages, Jaworki I Theil and Jaworki II Theil; \*\* - houses for Milówka, also covered Sucha Góra; \*\*\* - number of houses for Ostrów, which covered the villages of Ostrów and Rusiłów + additional brickyard building – Z.O. (ger. *Ziegelofen*)**


| | Vectorization (1861/62) | | Census (1869) | Map frame (1861/62) | | | |
|---|---|---|---|---|---|---|---|
| | Houses | Farm-related buildings | Houses | Houses | Stables | Number of people that can be housed | Number of horses that can be housed |
| **Examples of underestimation** | | | | | | | |
| **Jaworki** | 98 | 135 | 139 | 135\* | - | 30 | - |
| **Milcza** | 37 | 12 | 86 | 103 | 2 | 80 | 20 |
| **Examples of overestimation** | | | | | | | |
| **Milówka** | 281 | 184 | 279 | 230 + 30\*\*=260 | 6 | 150+10=160 | 14 |
| **Trzebinia** | 118 | 12 | 68 | 56 | - | - | - |



**Appendix A: Explanation on coordinate system terms, abbreviations and respective EPSG Geodetic Parameter Dataset codes.**

1942 coordinate system – the Soviet zonal projected coordinate system based on the Gauss-Krüger projection and Pulkovo-42 reference frame; used for military purposes in Warsaw Treaty countries till beginning of 1990s; Soviet military maps used in this project were elaborated in two of 6-degree zones – zone 4 (EPSG: 28404) and zone 5 (EPSG: 28405)

  1965 coordinate system – Polish official zonal projected coordinate system for large-scale topographic, civil maps used since 1968 till 2009; maps used in this project were elaborated in two zones – zone I, based on double stereographic projection and

460   Pulkovo-42 reference frame (EPSG: 3120) and zone V, based on Gauss-Krüger projection and Pulkovo-42 reference frame (EPSG: 2175)

  PL-1992 coordinate system – Polish official projected coordinate system used since beginning of 1990s for maps in scales 1:10 000 and lower (EPSG: 2180)

  ETRF-89 reference frame – European Terrestrial Reference Frame 1989, geodetic reference frame fixed to the stable part of

465   the Eurasian continental plate at epoch 1989.0 (EPSG: 1178)

  LAEA coordinate system – projected coordinate system based on Lambert Azimuthal Equal-Area projection (EPSG:3035).

  Pulkovo-42 reference frame – Soviet geodetic reference frame, based on Krasovsky 1940 ellipsoid, used in Warsaw Treaty countries, in Poland till 2009 (EPSG: 4284)

  UTM coordinate system – zonal projected coordinate system based on specific case of the transverse Mercator projection

470   called Universal Transverse Mercator; two of 6-degree UTM zones were used for the Ukrainian part of Galicia – zone 34N (EPSG: 32634) and zone 35N (EPSG: 32635)