# Peer review of "Mid-19th-century building structure locations in Galicia and Austrian Silesia under the Habsburg Monarchy"

_Earth System Science Data, 2020_

## Referee Comment (RC1) · Anonymous Referee #1 · 8 Jan 2021

This contribution describes the data set of building structures in the mid 19th century under which time the study area was under Austro-Hungarian rule. It is a very interesting capture of residential and (agricultural) outbuildings, from a time before accelerated human impact on landscape. The authors use of Second Military Survey maps from 1837-41 and 1861-64 as well as cadastral maps and census data ensures a high quality and gave the opportunity to cross check sources.

The methodology and approach is well explained and issues (such as accuracy for example) were anticipated and addressed in design. Section 3.4 is especially interesting in addressing local differences and discussing reasons behind this.

[Figure]

However, while acknowledging the two categories of buildings (First paragraph 2.2.4) this contribution would benefit from a brief explanation as to what other (non-agricultural) buildings are examples for the black category ("Wirtschaftsgebäude").

Technical Comments: Line 210: . . .20th century... ("XX century" does not conform with previous style) Line 240: . . .mid 19th century. . . ("mid-XIX century" does not conform with previous style) The following sources are not listed in the references: Stephens et al., 2019 (Line 16) Gingrich et al., 2019 (Line 19) Jepsen et al., 2015 (Line 19) Gavier-Pizarro et al., 2010 (Line 26) Gimmi et al., 2013 (Line 26/27) Kaim et al., 2018 (line 27) Affek, 2015 (Line 58) Skalos et al., 2011 (Line 58) Timár et al., 2010 (Line 58) Munteanu et al., 2015 (Line 60; Perhaps a typo for Munteanu et al., 2014) Feurdean et al., 2017 (Line 60) Pavelková et al., 2016 (Line 60)
* * *

---

## Referee Comment (RC2) · Anonymous Referee #2 · 17 Jan 2021

General comments:

This manuscript describes a vector dataset of reconstructed mid-19th-century building structure locations in former Galicia and Austrian Silesia covering an area of more than 80,000 km2 in present-day Czechia, Poland and Ukraine, derived from detailed Second Military Survey maps (at a scale of 1:28,800) that were built off of cadastral mapping (1:2,880) of the 19th century. The dataset includes two building categories, residential and outbuildings (mainly farming). The dataset is compared to census and cadastral data to evaluate local variations in differences between these and the extracted building data. The dataset is a useful resource that will be welcomed by researchers interested

in historical assessments of settlement, population and land use changes. The data represent the build structures in this regions at a very important point in time providing opportunities to better understand the evolution of the built environment and land use patterns over extended time periods. There are some concerns with this study and its design and the authors are encouraged to address them and add important detail and expand the scope of the research.

Specific comments:

There are three major issues. First, there is a significant lack of methods details. The authors dedicate no more than one sentence to the actual classification approach: "We used a semiautomatic, colour-based method involving the classification toolbar from ArcMap software." While the signature of the buildings might allow to use default tools to extract these symbols with high accuracy, the method underlying this ArcMap tool needs to be explained in detail. If there is not detail available it might not be the best idea to use a black box tool, to be frank. However, assuming, details can be found, the authors need to describe the underlying method/ type of classification run, parameters and any other aspects that might be relevant. The authors also need to ensure all details are included related to what they call "Data cleaning" in their workflow figure. Please make sure you include all the details necessary for any user to fully reproduce the methods and approaches and understand the choices made. Second, the validation of the classification results needs to be strengthened. It appears that the authors are validating the classification results for 1.3 Mio buildings using a sample of 1,500-1,600 objects. This is a 0.12% sample if this is all correctly understood. This represents a problem in terms of robustness and statistical power. This is true, especially as this validation is supposed to be valid across several dozens of map sheets that can be expected to have high levels of variation in their graphical properties and quality and thus, likely, the level of performance of the classification. The authors need to increase the sample size and based on underlying results from different map sheets show whether their validation statistics are representative and robust against underlying variation of the map images. This will make this validation step more credible for the data user. Also, a relative error measure would be a valuable addition to better understand the nature and magnitude of existing errors. Third, the authors need to think about ways to integrate uncertainty-related information in the final data product, and provide respective metadata that users can refer to for any quality-related aspects. There is no description entry (metadata) provided with the shapefile posted online. Uncertainty details will improve the data usefulness and instruct users about the fitness of the data for the intended use. This could include summaries of deviation statistics between the created data and the information on the map frame or the census-based data. Releasing such uncertainty-related information will increase the usability of and confidence in the data. The authors are encouraged to be creative on how this kind of information could be provided. It could be included in additional map-level files or for different regions.

The existing variation in agreements between the building data and the map frame information as well as the census data are very interesting. The authors are encouraged to add more of this exploration into the analysis of underlying uncertainties as they might be able to pave the way for some interesting substantive research on historical aspects of mapping and settlement patterns in the 19th century. For example, variation in such agreements could illustrate the role of other ancillary variables such as topography, water, transportation and accessibility. Such aspects would make the analysis of local differences much more interesting and provide more detail that users of the data could refer to in their applications.

Finally, it would be a valuable addition in the concluding part to lay out more detailed potential applications of the data to illustrate possible directions where it could be useful and which research areas could benefit by exploring new questions. To enrich the study, the authors could even consider the calculation of settlement change estimates using respective contemporary building data (or data layers that offer similar enough data such as the GHSL or the GUF data).

---

## Referee Comment (RC3) · Anonymous Referee #3 · 24 Jan 2021

The article "Mid-19th-century building structure locations in Galicia and Austrian Silesia under the Habsburg Monarchy" tries to reconstruct buildings locations in Galicia and Austrian Silesia in the period stated in the title. It brings a lot of new information based on the archival research of censuses data and analysis of cadastral and military maps. Although the manuscript in its present form is very interesting and informative, I recommend some changes. First of all, it should be explained in the introduction why exactly those two Habsburg provinces were chosen for analysis. I suppose the obvious reason is that part of both are today part of Poland. Perhaps it would be much better if authors concentrated only on Galicia, or if they compared (if there are) differences between those two provinces of the Habsburg/Austro-Hungarian Monarchy. Second,

it is not clear is your analysis covering only rural areas? If yes, it should be stated in the title. Third, it should be clearly explained what types of buildings are included. This is the biggest problem of this article, according to my opinion. The authors divide the buildings into two categories – "residential" and "outbuildings". However, what kind of buildings are those called "residential" is not clear, because, at the page 4 of the article it is stated that this category includes also "some churches, monasteries, town halls or railway stations". According to my opinion, it is not appropriate, because those are public and religious, and not residential buildings. It is also very weird that only party of them ("some") and not all of them are analyzed. If I did not understand properly, and if the authors did include all of religious and public buildings into their research then it should be clearly stated in the article. If not, they should change the title of the article so to emphasize that they analyze only residential and farm buildings. Are all of those buildings really called just Wohngebäude in archival sources? If yes, it seems rather unusual to me, considering that Austrian surveys were mostly very precise. To conclude – if not all of those buildings were residential, then you cannot call them residential. Furthermore, although public and sacral buildings comprise only 1% of the buildings marked with "red" on maps, they were, almost always, the biggest buildings in places, so they should be included into your research. This way your article would be much useful for historians of architecture too. Regarding the second type of buildings that authors are analyzing, the term that they use - outbuildings – is very unusual, at least in architectural history. If I understood properly, it is the translation of the German word Wirtschaftsgebäude, and the authors also use for this type of buildings term "farm buildings". This German word, however, has broader meaning – Wirtschaftsgebäude are not only farm buildings, as can be clearly seen from dictionaries. Fourth and the last thing: on the page 6 it is stated "The censuses closest in time to the publication of the maps were organized in 1857 for Austrian Silesia (n=23) and in 1869 for Galicia (n=76)". According to my knowledge, both censuses (in 1857 and 1869) were organized in the whole Habsburg Monarchy, therefore also in Silesia and Galicia on both occasions.

---

## Author Comment (AC1) · 2 Mar 2021

Response to referees comments to the preprint essd-2020-379

| # | Comment | Response |
|---|---------|----------|
| | **Reviewer 1 (posted on 08 Jan 2021)** | |
| 1. | This contribution describes the data set of building structures in the mid 19th century under which time the study area was under Austro-Hungarian rule. It is a very interesting capture of residential and (agricultural) outbuildings, from a time before accelerated human impact on landscape. The authors use of Second Military Survey maps from 1837-41 and 1861-64 as well as cadastral maps and census data ensures a high quality and gave the opportunity to cross check sources.
The methodology and approach is well explained and issues (such as accuracy for example) were anticipated and addressed in design. Section 3.4 is especially interesting in addressing local differences and discussing reasons behind this. | Thank you! |
| 2. | However, while acknowledging the two categories of buildings (First paragraph 2.2.4) this contribution would benefit from a brief explanation as to what other (nonagricultural) buildings are examples for the black category ("Wirtschaftsgebäude"). | We decided to add a new figure (Figure 3) showing the examples of non-residential, although marked with red, examples of buildings (A – monastery, B – church), and unusual buildings marked with black (C – stone- and brick-made sheepfold, D – railway station). Please note that the 'Castle' as seen in example C, refers to the neighboring building marked with red. Among the buildings marked with black, we found also occasionally some chapels. One can say, that potentially in such cases, the black building might have been an indication of wood, as a building material, but we found many examples showing that it is not the case. Because of the scale of the work (> 450 map sheets, > 80 000 km$^2$), we are not able to explain all the exceptions and also, we have limited options to use independent sources on the local level, as a validation source. That is why, we decided in this paper to focus on the two main building categories, according to the map legend (marked with red and black), but plan also in future to go deeper in the text and signature analysis of the map content in order to present in detail, the above-mentioned 2% of the structures to the scientific community. |
| 3. | Technical Comments: Line 210: : : :20th century... ("XX century" does not conform with previous style) Line 240: : : :mid 19th century: : : ("mid-XIX century" does not conform with previous style) | Thank you, corrected. |
| 4. | The following sources are not listed in the references: Stephens et al., 2019 (Line 16) Gingrich et al., 2019 (Line 19) Jepsen et al., 2015 (Line 19) Gavier- Pizarro et al., 2010 (Line 26) Gimmi et al., 2013 (Line 26/27) Kaim et al., 2018 (line 27) Affek, | All the respective references were added and the quality of the presentation of the reference list was improved. |

| | | |
|---|---|---|
| | 2015 (Line 58) Skalos et al., 2011 (Line 58) Timár et al., 2010 (Line 58) Munteanu et al., 2015 (Line 60; Perhaps a typo for Munteanu et al., 2014) Feurdean et al., 2017 (Line 60) Pavelková et al., 2016 (Line 60) | |
| | | |
| | **Reviewer 2 (posted on 17 Jan 2021)** | |
| 1. | General comments: This manuscript describes a vector dataset of reconstructed mid-19th-century building structure locations in former Galicia and Austrian Silesia covering an area of more than 80,000 km2 in present-day Czechia, Poland and Ukraine, derived from detailed Second Military Survey maps (at a scale of 1:28,800) that were built off of cadastral mapping (1:2,880) of the 19th century. The dataset includes two building categories, residential and outbuildings (mainly farming). The dataset is compared to census and cadastral data to evaluate local variations in differences between these and the extracted building data. The dataset is a useful resource that will be welcomed by researchers interested in historical assessments of settlement, population and land use changes. The data represent the build structures in this regions at a very important point in time providing opportunities to better understand the evolution of the built environment and land use patterns over extended time periods. There are some concerns with this study and its design and the authors are encouraged to address them and add important detail and expand the scope of the research. | Thank you for the kind words. We improved the manuscript according to the suggestions and present it in details below. |
| 2 | Specific comments: There are three major issues. First, there is a significant lack of methods details. The authors dedicate no more than one sentence to the actual classification approach: "We used a semiautomatic, colour-based method involving the classification toolbar from ArcMap software." While the signature of the buildings might allow to use default tools to extract these symbols with high accuracy, the method underlying this ArcMap tool needs to be explained in detail. If there is not detail available it might not be the best idea to use a black box tool, to be frank. However, assuming, details can be found, the authors need to describe the underlying method/ type of classification run, parameters and any other aspects that might be relevant. The authors also need to ensure all details are included related to what they call "Data cleaning" in their workflow figure. Please make sure you include all the details | The section was substantially improved by adding the details about parameters we used, mainly threshold values for the size and shape of the objects. However, taking into account that there were substantial differences in the map sheet quality, we explain that the color-based, initial classification had to be repeated on a separate set of training data several times. Additionally, a relatively high quality of the final dataset was achieved thanks to the manual verification of each of the > 450 map sheets. It is also important to add that we were not primarily focused on the creation of the universal method to acquire buildings from historical maps, but rather wanted to employ a set of rules fitted to our conditions and finally, useful enough to help us in relatively fast structures acquisition. Overall, we hope that our procedure might be helpful for other scholars in creating their own classifications, and on the other |

| | | |
|---|---|---|
| | necessary for any user to fully reproduce the methods and approaches and understand the choices made. | hand, is clear enough for the readers, explaining how we collected objects in our database. |
| 3 | Second, the validation of the classification results needs to be strengthened. It appears that the authors are validating the classification results for 1.3 Mio buildings using a sample of 1,500-1,600 objects. This is a 0.12% sample if this is all correctly understood. This represents a problem in terms of robustness and statistical power. This is true, especially as this validation is supposed to be valid across several dozens of map sheets that can be expected to have high levels of variation in their graphical properties and quality and thus, likely, the level of performance of the classification. The authors need to increase the sample size and based on underlying results from different map sheets show whether their validation statistics are representative and robust against underlying variation of the map images. This will make this validation step more credible for the data user. Also, a relative error measure would be a valuable addition to better understand the nature and magnitude of existing errors. | According to the suggestion, we strengthen the analysis, by starting with 1000, instead of 300, randomly selected circles (300 m ratio; area – 28.27 ha), where we first checked if there are any buildings in the database or on the map. Because of that, the final number of test circles was reduced to 311, containing 4791 buildings (previously 93 circles). Based on that sample, we calculated the margin of error on 1.86% (confidence level – 99%, population size – 1,305,233). The results of the procedure are included in the revised version of the manuscript (e.g. overall accuracy was improved when compared to the previous sample from 93.65% to 95.03%). The variations among randomly selected circles, located on different map sheets can be found in Figure 5, where Pearson's correlations between the number of buildings shown on the maps and the number of structures acquired in the dataset can be consulted. Additionally in the revised version of the figure, we added also locations of the 311 randomly selected test areas, to show that they represent the entire study area. Overall, we hope that now the procedure makes it possible to conclude on the quality of our database. Additionally, apart from the visually assessed database quality presented above, we verified it also using other independent sources like census data and map frame information, where the sample sizes were substantially higher. It is included in the manuscript, as it was already in the previous version. |
| 4 | Third, the authors need to think about ways to integrate uncertainty-related information in the final data product, and provide respective metadata that users can refer to for any quality-related aspects. There is no description entry (metadata) provided with the shapefile posted online. Uncertainty details will improve the data usefulness and instruct users about the fitness of the data for the intended use. This could include summaries of deviation statistics between the created data and the information on the map frame or the census-based data. Releasing such uncertainty-related information will increase the usability of and confidence in the data. The authors are encouraged to be creative on how this kind of information could be provided. It could be included in additional map-level files or for different regions. | We decided to add a separate polygon layer of districts covering the entire study area, where we added the attributes including: year of the census, year of map creation (dominating value for the district unit), time difference between map and census dates, as well as number of houses according to the census and according to the database, and finally the percentage of the residential structures in the database in relation to census data. We hope that such an auxiliary dataset will help in defining potential uncertainties responsible for differences found in the data. Respective clarifications were added to the manuscript. Explanations on attribute names and content can be found in *Data availability* section. |
| 5 | The existing variation in agreements between the building data and the map frame information as well as the census data are very interesting. The authors are encouraged to add more of this exploration into the analysis of underlying uncertainties as they | Apart from the metadata presented above, we decided to include also other spatial determinants, which might, at least partly, influence the deviations between the mapped data and census data. In order to make it clear to the readers and data users, we included |

| | | |
|---|---|---|
| | might be able to pave the way for some interesting substantive research on historical aspects of mapping and settlement patterns in the 19th century. For example, variation in such agreements could illustrate the role of other ancillary variables such as topography, water, transportation and accessibility. Such aspects would make the analysis of local differences much more interesting and provide more detail that users of the data could refer to in their applications. | Appendix B, where we present a set of variables on the district level. The maps show: number of houses in the database, as a percentage of census records of homes, time difference between map, and census publication, population density, mean distance to main roads, mean elevation and mean slope. All the data can be found in the attributes attached to the shapefile with the district map mentioned in the previous comment and all the respective explanations showing how we acquired the variables, are added to the manuscript. Apart from presenting raw data, we did also a correlation analysis, where deviations between the mapped data and census data were checked against the above-mentioned variables. Unfortunately, the only correlation, which was statistically significant ($p < 0.05$), was the correlation with the time difference between map and census publication – $r = 0.217$. Unfortunately, preparation of such analysis, based on map frame information details is not possible for large areas, as it is based on the comparison of data on the village level. As we already mentioned in section 3.3. in the manuscript, in many cases, the villages were split into neighboring map sheets, and corrections, including adding or removing some buildings located within the specified villages, have to be implemented at this level of analysis. That is why we decided to present the analysis on the agreement to the whole study area, however, based on the census data. Comparison to the map frame information remained the same, as in the previous version of the manuscript. |
| 6 | Finally, it would be a valuable addition in the concluding part to lay out more detailed potential applications of the data to illustrate possible directions where it could be useful and which research areas could benefit by exploring new questions. To enrich the study, the authors could even consider the calculation of settlement change estimates using respective contemporary building data (or data layers that offer similar enough data such as the GHSL or the GUF data). | We added a paragraph showing the potential applications to the *Conclusions*, as suggested. However, we decided not to enlarge the Data Descriptor, by adding settlement change estimates, based on current data. First, we think that it would require a lot of changes in the manuscript, incl. Methods, Results and Discussion, what would negatively affect the focus of the paper. Second, the ESSD requirements state that detailed analysis as might be reported in a research article (and we think that such comparison might have a form of regular analysis) remain outside the scope of this data journal. |
| | | |
| | **Reviewer 3 (posted on 24 Jan 2021)** | |
| 1 | The article "Mid-19th-century building structure locations in Galicia and Austrian Silesia under the Habsburg Monarchy" tries to reconstruct buildings locations in Galicia and Austrian Silesia in the period stated in the title. It brings a lot of new information based on the archival research of censuses data and analysis of cadastral and military | Thank you for the kind words. The manuscript was improved according to the suggestions of three Reviewers. The details are presented next to each comment. |

| | | |
|---|---|---|
| | maps. Although the manuscript in its present form is very interesting and informative, I recommend some changes. | |
| 2 | First of all, it should be explained in the introduction why exactly those two Habsburg provinces were chosen for analysis. I suppose the obvious reason is that part of both are today part of Poland. Perhaps it would be much better if authors concentrated only on Galicia, or if they compared (if there are) differences between those two provinces of the Habsburg/Austro-Hungarian Monarchy. | We explain briefly the context in the Study area section, where we added the information on close linkages between the regions due to economic and social reasons, which makes showing them together a rational choice. However, the reason mentioned by the Reviewer also plays important role in study area defining. This work is a part of larger project, where both provinces are studied in detail (respective clarification can be found in Acknowledgments), and the fact that areas are today part of Poland was one of the reason to study it in that form. |
| 3 | Second, it is not clear is your analysis covering only rural areas? If yes, it should be stated in the title. | Our analysis covers both rural and urban areas. The buildings in towns were also vectorized and compared to the census data. In the revised version of the manuscript, we added also the shape, polygon layer with a set of variables presented on the district level, which shows also urban districts and the level of deviations between the database and census. It is important to note, however, that a substantial part of the provinces was rural indeed, and located in mountainous regions of the Carpathians and Sudety Mountains. |
| 4. | Third, it should be clearly explained what types of buildings are included. This is the biggest problem of this article, according to my opinion. The authors divide the buildings into two categories – "residential" and "outbuildings". However, what kind of buildings are those called "residential" is not clear, because, at the page 4 of the article it is stated that this category includes also "some churches, monasteries, town halls or railway stations". According to my opinion, it is not appropriate, because those are public and religious, and not residential buildings. It is also very weird that only party of them ("some") and not all of them are analyzed. If I did not understand properly, and if the authors did include all of religious and public buildings into their research then it should be clearly stated in the article. If not, they should change the title of the article so to emphasize that they analyze only residential and farm buildings. Are all of those buildings really called just Wohngebäude in archival sources? If yes, it seems rather unusual to me, considering that Austrian surveys were mostly very precise. To conclude – if not all of those buildings were residential, then you cannot call them residential. Furthermore, although public and sacral buildings comprise only 1% of the buildings marked with "red" on maps, they were, almost always, the biggest buildings in places, so | Our aim was to present the buildings in line with the original source data – the Second Military Survey. Since the original instruction to the maps is not available, we based our work on the publication of Zaffauk (1889), which presents the symbols shown on the map. It seems that the main division was between the residential buildings (ger. Wohngebäude) and farm-related buildings (ger. Wirtschaftsgebäude). However, in order to better communicate the exceptions we encountered, we decided to add also new figure (Figure 3), where we show what also may be found among residential being actually non-residential (A – monastery, B – church) or what was marked with black, not being farm-related (e.g. D – railway station). Among the buildings marked with black, we found also occasionally some chapels. One can say, that potentially in such cases, the black buildings might have been an indication of wood, as a building material, but we found many examples confirming that it was not the case. The Second Military Survey contained also the textual information and signatures indicating different types of buildings (incl. churches, chapels, monasteries or mills), but it was somehow independent of the basic division on red and black buildings presented above. Very often the text or signature is not easily combined to specified structure, but rather to the proximate location (as e.g. in Figure 3, where 'Castle' as seen in the example, |

| | they should be included into your research. This way your article would be much useful for historians of architecture too. | refers not to the closest, black structure, but to the neighboring building marked with red). We are currently working on this specific information indicating building functions, but since it requires other sources and methods of validation, we decided not to include it in this paper. Here we decided to stay with the basic division only, to be in line with the map legend. Please note that the exceptions refer to ~ 1% of the objects only, which was also confirmed by comparing a number of houses in our database to the number of houses recorded in the census. We hope that in-depth studies of functional buildings will be ready soon to share with the wider community. It is also important to add, that since the map was prepared for military purposes, some of the buildings were not marked as not important from the military point of view (e.g. synagogues), while others were included (churches, usually with towers), as potentially important from the orientational point of view, so the image of buildings included is strongly related to the aim of the map. |
|---|---|---|
| 5 | Regarding the second type of buildings that authors are analyzing, the term that they use - outbuildings – is very unusual, at least in architectural history. If I understood properly, it is the translation of the German word Wirtschaftsgebäude, and the authors also use for this type of buildings term "farm buildings". This German word, however, has broader meaning – Wirtschaftsgebäude are not only farm buildings, as can be clearly seen from dictionaries. | As mentioned above, we based our analysis on the publication of Zaffauk (1889), who explains the map symbols and the term 'Wirtschaftsgebäude' appears there. We wanted to be in line with the map legend, so as the building was black, we mark it in that way. It is clearly defined in our database. Other, very often interesting analyses, require more detailed, often very local sources of validation, which was beyond the scope of analysis taking into account the area under study (> 80 000 km$^2$), and a number of map sheets (> 450), we processed. However, we hope that the explanations we did and the new Figure 3, we have added in the revised version of the manuscript will show the potential users, that not all of the buildings marked with black are actually farm-related, as noted by the Reviewer. |
| 6 | Fourth and the last thing: on the page 6 it is stated "The censuses closest in time to the publication of the maps were organized in 1857 for Austrian Silesia (n=23) and in 1869 for Galicia (n=76)". According to my knowledge, both censuses (in 1857 and 1869) were organized in the whole Habsburg Monarchy, therefore also in Silesia and Galicia on both occasions. | The difference we had to cope with was the map creation period – 1837–1841 for Austrian Silesia and 1861–1864 for Galicia. The censuses closest in time to the publication of the maps were organized in 1857 for Austrian Silesia and in 1869 for Galicia. That is why we decided to use different census for each of the regions. We decided to add an additional shapefile layer with metadata on the district level (see comment 4 of Reviewer 2), helping in defining how it might impact the differences in numbers of structures between our database and the census. Some of the additional metadata are also shown as the maps in Appendix B. |

---

## Author Response (AR1)

*To the Editor of*
*Earth System Science Data*

*Dr. Kirsten Elger*

March 8[th] 2021

Dear Dr. Elger

On behalf of all co-authors, I am pleased to present to you our revised version of the data descriptor "**Mid-19th-century building structure locations in Galicia and Austrian Silesia under the Habsburg Monarchy**" for potential publication as a paper in Earth System Science Data.

We improved the manuscript substantially, according to the Reviewer's comments (attached as a separate document below). The main aspects we changed cover expanding the accuracy assessment test areas from 93 to 311 and adding a margin of error value, adding a new set of uncertainty-related data, as the separate attributes in new polygon layer (they cover entire study area). Minor remarks raised by the Reviewers were also addressed. Additionally, we attach the manuscript with tracked changes, where our improvements (marked in blue) and some minor English proofreading changes (in red) are highlighted. The changes in red are proposed by the professional English editor and we accepted them in revised version of the manuscript. We hope that now the manuscript has higher potential and have a chance to be published in ESSD.

We look forward to hearing from you.

Sincerely,

Dominik Kaim

Responses to the referees' comments to the preprint essd-2020-379

| # | Comment | Response | Lines |
|---|---------|----------|-------|
| | **Reviewer 1 (posted on 08 Jan 2021)** | | |
| 1. | This contribution describes the data set of building structures in the mid 19th century under which time the study area was under Austro-Hungarian rule. It is a very interesting capture of residential and (agricultural) outbuildings, from a time before accelerated human impact on landscape. The authors use of Second Military Survey maps from 1837-41 and 1861-64 as well as cadastral maps and census data ensures a high quality and gave the opportunity to cross check sources.
The methodology and approach is well explained and issues (such as accuracy for example) were anticipated and addressed in design. Section 3.4 is especially interesting in addressing local differences and discussing reasons behind this. | Thank you! | |
| 2. | However, while acknowledging the two categories of buildings (First paragraph 2.2.4) this contribution would benefit from a brief explanation as to what other (nonagricultural) buildings are examples for the black category ("Wirtschaftsgebäude"). | We added a new figure (Figure 3) that shows the examples of non-residential buildings marked in red, which are examples of regular buildings (A – monastery, B – church), and unusual buildings marked in black (C – stone- and brick-made sheepfold, D – railway station). Please note that the 'Castle' as seen in example C refers to the neighbouring building marked in red. Among the buildings marked in black, we also occasionally found some chapels. Potentially, in such cases, the black building may have been an indication of wood as a building material, but we found many examples that show that this is not the case. Because of the scale of the work (> 450 map sheets, > 80 000 km$^2$), we are not able to explain all the exceptions, and we also have limited options to use independent sources at the local level as a validation source. For this reason, in this paper, we focused on the two main building categories according to the map legend (marked in red and black), but we also plan in the future to more deeply examine the text and signature analysis of the map content to present in detail the abovementioned 2% of the structures to the scientific community. | 510 |
| 3. | Technical Comments: Line 210: : : :20th century... ("XX century" does not conform with previous style) Line 240: : : :mid 19th century: : | Thank you. This has been corrected. | |

| | | | |
|---|---|---|---|
| | : ("mid-XIX century" does not conform with previous style) | | |
| 4. | The following sources are not listed in the references: Stephens et al., 2019 (Line 16) Gingrich et al., 2019 (Line 19) Jepsen et al., 2015 (Line 19) Gavier- Pizarro et al., 2010 (Line 26) Gimmi et al., 2013 (Line 26/27) Kaim et al., 2018 (line 27) Affek, 2015 (Line 58) Skalos et al., 2011 (Line 58) Timár et al., 2010 (Line 58) Munteanu et al., 2015 (Line 60; Perhaps a typo for Munteanu et al., 2014) Feurdean et al., 2017 (Line 60) Pavelková et al., 2016 (Line 60) | All the respective references were added, and the quality of the presentation of the reference list was improved. | |
| | | | |
| | **Reviewer 2 (posted on 17 Jan 2021)** | | |
| 1. | General comments: This manuscript describes a vector dataset of reconstructed mid-19th-century building structure locations in former Galicia and Austrian Silesia covering an area of more than 80,000 km2 in present-day Czechia, Poland and Ukraine, derived from detailed Second Military Survey maps (at a scale of 1:28,800) that were built off of cadastral mapping (1:2,880) of the 19th century. The dataset includes two building categories, residential and outbuildings (mainly farming). The dataset is compared to census and cadastral data to evaluate local variations in differences between these and the extracted building data. The dataset is a useful resource that will be welcomed by researchers interested in historical assessments of settlement, population and land use changes. The data represent the build structures in this regions at a very important point in time providing opportunities to better understand the evolution of the built environment and land use patterns over extended time periods. There are some concerns with this study and its design and the authors are encouraged to address them and add important detail and expand the scope of the research. | Thank you for your kind words. We improved the manuscript according to the suggestions, and present our improvements in detail below. | |
| 2 | Specific comments: There are three major issues. First, there is a significant lack of methods details. The authors dedicate no more than one sentence to the actual classification approach: "We used a semiautomatic, colour-based method involving the classification toolbar from ArcMap software." While the signature of the buildings | The section was substantially improved by adding details about the parameters that we used, which were mainly the threshold values for the size and shape of the objects. However, taking into account that there were substantial differences in the map sheet quality, we explain that the initial colour-based classification had to be repeated on a separate set of training data | 117-136 |

| | | | |
|---|---|---|---|
| | might allow to use default tools to extract these symbols with high accuracy, the method underlying this ArcMap tool needs to be explained in detail. If there is not detail available it might not be the best idea to use a black box tool, to be frank. However, assuming, details can be found, the authors need to describe the underlying method/ type of classification run, parameters and any other aspects that might be relevant. The authors also need to ensure all details are included related to what they call "Data cleaning" in their workflow figure. Please make sure you include all the details necessary for any user to fully reproduce the methods and approaches and understand the choices made. | several times. Additionally, a relatively high quality of the final dataset was achieved because of the manual verification of each of the > 450 map sheets. It is also important to add that we were not primarily focused on the creation of the universal method to acquire buildings from historical maps but rather wanted to employ a set of rules that fit with our conditions and that were finally, sufficiently useful to help us in a relatively rapid structure acquisition. Overall, we hope that our procedure not only is helpful to other scholars in creating their own classifications but also is clear enough for the readers by explaining how we collected the objects in our database. | |
| 3 | Second, the validation of the classification results needs to be strengthened. It appears that the authors are validating the classification results for 1.3 Mio buildings using a sample of 1,500-1,600 objects. This is a 0.12% sample if this is all correctly understood. This represents a problem in terms of robustness and statistical power. This is true, especially as this validation is supposed to be valid across several dozens of map sheets that can be expected to have high levels of variation in their graphical properties and quality and thus, likely, the level of performance of the classification. The authors need to increase the sample size and based on underlying results from different map sheets show whether their validation statistics are representative and robust against underlying variation of the map images. This will make this validation step more credible for the data user. Also, a relative error measure would be a valuable addition to better understand the nature and magnitude of existing errors. | According to the suggestion, we strengthened the analysis by starting with 1,000 instead of 300 randomly selected circles (300-m ratio; area – 28.27 ha), where we first checked if there were any buildings in the database or on the map. Accordingly, the final number of test circles was reduced to 311, which contained 4,791 buildings (previously 93 circles). Based on this sample, we calculated the margin of error on 1.86% (confidence level – 99%, population size – 1,305,233). The results of the procedure are included in the revised version of the manuscript (e.g., the overall accuracy was improved compared to the previous sample from 93.65% to 95.03%). The variations among the randomly selected circles located on different map sheets can be found in Figure 5, where the Pearson's correlations between the number of buildings shown on the maps and the number of structures acquired from the dataset can be consulted. Additionally, in the revised version of the figure, we also added the locations of the 311 randomly selected test areas to show that they represent the entire study area. Overall, we hope that the procedure now makes it possible to make a conclusion on the quality of our database.

Furthermore, apart from the visually assessed database quality presented above, we verified it by also using other independent sources such as census data and map frame information, where the sample sizes were substantially higher. This information is included in the manuscript, as it was already in the previous version. | 144-172 |
| 4 | Third, the authors need to think about ways to integrate uncertainty-related information in the final data product, and provide respective metadata that users can refer to for any quality- | We added a separate polygon layer of districts that covers the entire study area, where we added the attributes including the year of the census, year of map creation (the dominating | 180-187 |

| | | |
|---|---|---|
| | related aspects. There is no description entry (metadata) provided with the shapefile posted online. Uncertainty details will improve the data usefulness and instruct users about the fitness of the data for the intended use. This could include summaries of deviation statistics between the created data and the information on the map frame or the census-based data. Releasing such uncertainty-related information will increase the usability of and confidence in the data. The authors are encouraged to be creative on how this kind of information could be provided. It could be included in additional map-level files or for different regions. | value for the district unit), time difference between the map and census dates, number of houses according to the census and according to the database, and finally, percentage of the residential structures in the database in relation to the census data. We hope that such an auxiliary dataset will help in identifying the potential uncertainties responsible for the differences found in the data. The respective clarifications were added to the manuscript. The definitions of the attributes can be found in the *Data availability* section. | |
| 5 | The existing variation in agreements between the building data and the map frame information as well as the census data are very interesting. The authors are encouraged to add more of this exploration into the analysis of underlying uncertainties as they might be able to pave the way for some interesting substantive research on historical aspects of mapping and settlement patterns in the 19th century. For example, variation in such agreements could illustrate the role of other ancillary variables such as topography, water, transportation and accessibility. Such aspects would make the analysis of local differences much more interesting and provide more detail that users of the data could refer to in their applications. | Apart from the metadata presented above, we also included other spatial determinants, which might at least partly influence the deviations between the mapped data and census data. To make this clear to the readers and data users, we included Appendix B, where we present a set of variables at the district level. The maps show the number of houses in the database as a percentage of the census records of homes, time difference between map and census publication, population density, mean distance to main roads, mean elevation and mean slope. All the data can be found in the attributes attached to the shapefile with the district map mentioned in the previous comment, and all the respective explanations that show how we acquired the variables were added to the manuscript. Apart from presenting raw data, we also conducted a correlation analysis, where the deviations between the mapped data and census data were checked against the abovementioned variables. Unfortunately, the only correlation that was statistically significant ($p < 0.05$) was the correlation with the time difference between the map and census publication – $r = 0.217$. Unfortunately, the preparation of such analysis based on map frame information details is not possible for large areas, as it is based on the comparison of data at the village level. As we mentioned in section 3.3. in the manuscript, in many cases, the villages were split into neighbouring map sheets, and corrections, including adding or removing some buildings located within the specified villages, have to be implemented at this level of analysis. For this reason, we instead presented an analysis on the agreement with the entire study area based on the census data. A comparison of these results to | Appendix B, 201-204 |

| | | the map frame information remained the same, as in the previous version of the manuscript. | |
|---|---|---|---|
| 6 | Finally, it would be a valuable addition in the concluding part to lay out more detailed potential applications of the data to illustrate possible directions where it could be useful and which research areas could benefit by exploring new questions. To enrich the study, the authors could even consider the calculation of settlement change estimates using respective contemporary building data (or data layers that offer similar enough data such as the GHSL or the GUF data). | We added a paragraph that shows the potential applications to the *Conclusion*, as suggested. However, we decided not to enlarge the Data Descriptor, by adding settlement change estimates, based on current data. First, we think that it would require a lot of changes to the manuscript, including the Methods, Results and Discussion sections, which would negatively affect the focus of the paper. Second, the ESSD requirements state that a detailed analysis, which might be reported in a research article (and we think that such comparison might have a form of regular analysis), remains outside the scope of this data journal. | 330-335 |
| | | | |
| | **Reviewer 3 (posted on 24 Jan 2021)** | | |
| 1 | The article "Mid-19th-century building structure locations in Galicia and Austrian Silesia under the Habsburg Monarchy" tries to reconstruct buildings locations in Galicia and Austrian Silesia in the period stated in the title. It brings a lot of new information based on the archival research of censuses data and analysis of cadastral and military maps. Although the manuscript in its present form is very interesting and informative, I recommend some changes. | Thank you for your kind words. The manuscript was improved according to the suggestions of three reviewers. The details are presented next to each comment. | |
| 2 | First of all, it should be explained in the introduction why exactly those two Habsburg provinces were chosen for analysis. I suppose the obvious reason is that part of both are today part of Poland. Perhaps it would be much better if authors concentrated only on Galicia, or if they compared (if there are) differences between those two provinces of the Habsburg/Austro-Hungarian Monarchy. | We briefly explain the context in the study area section, where we added the information on the close linkages between the regions due to economic and social reasons, which makes studying them together a rational choice. However, the reason mentioned by the reviewer also plays an important role in defining a study area. This work is a part of a larger project, where both provinces are studied in detail (the respective clarification can be found in Acknowledgments), and the fact that the areas are currently part of Poland was one of the reasons to study them in this form. | 47-49 |
| 3 | Second, it is not clear is your analysis covering only rural areas? If yes, it should be stated in the title. | Our analysis covers both rural and urban areas. The buildings in towns were also vectorized and compared to the census data. In the revised version of the manuscript, we also added the shape and the polygon layer with a set of variables presented at the district level, which also show urban districts and the level of the deviations between the database and the census. It is important to note, however, that a substantial part of the provinces was indeed | |

| | | rural and located in the mountainous regions of the Carpathians and Sudety Mountains. | |
|---|---|---|---|
| 4. | Third, it should be clearly explained what types of buildings are included. This is the biggest problem of this article, according to my opinion. The authors divide the buildings into two categories – "residential" and "outbuildings". However, what kind of buildings are those called "residential" is not clear, because, at the page 4 of the article it is stated that this category includes also "some churches, monasteries, town halls or railway stations". According to my opinion, it is not appropriate, because those are public and religious, and not residential buildings. It is also very weird that only party of them ("some") and not all of them are analyzed. If I did not understand properly, and if the authors did include all of religious and public buildings into their research then it should be clearly stated in the article. If not, they should change the title of the article so to emphasize that they analyze only residential and farm buildings. Are all of those buildings really called just Wohngebäude in archival sources? If yes, it seems rather unusual to me, considering that Austrian surveys were mostly very precise. To conclude – if not all of those buildings were residential, then you cannot call them residential. Furthermore, although public and sacral buildings comprise only 1% of the buildings marked with "red" on maps, they were, almost always, the biggest buildings in places, so they should be included into your research. This way your article would be much useful for historians of architecture too. | Our aim was to present the buildings in line with the original source data – the Second Military Survey. Since the original instruction to the maps is not available, we based our work on the publication of Zaffauk (1889), which presents the symbols shown on the map. The main division seems to be between the residential buildings (ger. Wohngebäude) and farm-related buildings (ger. Wirtschaftsgebäude). However, to better communicate the exceptions that we encountered, we added a new figure (Figure 3), where we show what also may be found among residential buildings that are actually non-residential (A – monastery, B – church) or what was marked in black not being farm-related (e.g., D – railway station). Among the buildings marked in black, we occasionally found some chapels. Potentially, in such cases, the black buildings might have been an indication of wood as a building material, but we found many examples that confirm that this was not the case. The Second Military Survey also contained the textual information and signatures that indicated different types of buildings (including churches, chapels, monasteries or mills), but it was somehow independent of the basic division on red and black buildings presented above. Very often, the text or signature was not easily combined with a specified structure but rather with the proximate location (e.g., in Figure 3, where 'Castle' as seen in the example refers not to the closest black structure but to the neighbouring building marked in red). We are currently working on this specific information that indicates building functions, but since it requires other sources and methods of validation, we did not include it in this paper. Here, we stayed only with the basic division to be consistent with the map legend. Please note that the exceptions refer to only ~1% of the objects, which was also confirmed by comparing a number of houses in our database to a number of houses recorded in the census. We hope that in-depth studies of functional buildings will soon be ready to share with the wider community. It is also important to add that since the map was prepared for military purposes, some of the buildings were not marked as they were not important from a military point of view (e.g., synagogues), while | 105-115, Figure 3 |

| | | others were included (churches, usually with towers) as potentially important from the orientational point of view; therefore, the images of the included buildings is strongly related to the aim of the map. | |
|---|---|---|---|
| 5 | Regarding the second type of buildings that authors are analyzing, the term that they use - outbuildings – is very unusual, at least in architectural history. If I understood properly, it is the translation of the German word Wirtschaftsgebäude, and the authors also use for this type of buildings term "farm buildings". This German word, however, has broader meaning – Wirtschaftsgebäude are not only farm buildings, as can be clearly seen from dictionaries. | As mentioned above, we based our analysis on the publication of Zaffauk (1889), who explains the map symbols, and the term 'Wirtschaftsgebäude' appears there. We wanted to be in line with the map legend; thus, as the building was black, we also mark it this way. This is clearly defined in our database. Other very often interesting analyses require more detailed and often very local sources of validation, which was beyond the scope of analysis by taking into account the area under study ($> 80\ 000\ km^2$) and the number of map sheets ($> 450$) that we processed. However, we hope that our explanations and the new Figure 3 that we have added to the revised version of the manuscript will show the potential users that not all of the buildings marked in black are actually farm-related, as noted by the reviewer. | 105-115, Figure 3 |
| 6 | Fourth and the last thing: on the page 6 it is stated "The censuses closest in time to the publication of the maps were organized in 1857 for Austrian Silesia (n=23) and in 1869 for Galicia (n=76)". According to my knowledge, both censuses (in 1857 and 1869) were organized in the whole Habsburg Monarchy, therefore also in Silesia and Galicia on both occasions. | The difference that we had to cope with was the map creation period – 1837–1841 for Austrian Silesia and 1861–1864 for Galicia. The censuses closest in time to the publication of the maps were organized in 1857 for Austrian Silesia and in 1869 for Galicia. For this reason, we used different censuses for each of the regions. We added an additional shapefile layer with metadata on the district level (see comment 4 of Reviewer 2), which helped in defining how it might impact the differences in the numbers of structures between our database and the census. Some of the additional metadata are also shown as maps in Appendix B. | 174-179 |